# Chinese sign language recognition based on surface electromyography and motion information

**Wenyu Li**[1], **Zhizeng Luo**[1]\*, **Wenguo Li**[1,2], **Xugang Xi**[1]

**1** Institute of Intelligent Control and Robotics, Hangzhou Dianzi University, Hangzhou, Zhejiang, China,
**2** Xianheng International (Hangzhou) Electric Manufacturing Co., Ltd., Hangzhou, Zhejiang, China

\* luo@hdu.edu.cn

## Abstract

Sign language (SL) has strong structural features. Various gestures and the complex trajectories of hand movements bring challenges to sign language recognition (SLR). Based on the inherent correlation between gesture and trajectory of SL action, SLR is organically divided into gesture-based recognition and gesture-related movement trajectory recognition. One hundred and twenty commonly used Chinese SL words involving 9 gestures and 8 movement trajectories, are selected as research and test objects. The method based on the amplitude state of surface electromyography (sEMG) signal and acceleration signal is used for vocabulary segmentation. The multi-sensor decision fusion method of coupled hidden Markov model is used to complete the recognition of SL vocabulary, and the average recognition rate is 90.41%. Experiments show that the method of sEMG signal and motion information fusion has good practicability in SLR.

## Introduction

Sign language (SL) is the main way for deaf/mute individuals to communicate, which enables them to improve their social participation. Sign language recognition (SLR) uses computers to convert the information expressed by SL actions into specific target application information, which has become one of the research hotspots in the field of rehabilitation medicine [1, 2].

Traditional SLR technologies based on images and data gloves do not easily meet the requirements of wearability and low cost [3, 4], and the recognition method based on the combination of surface electromyography (sEMG) and motion information is gradually favored [5–7]. Approximately 5600 types of Chinese sign language (CSL) vocabulary exist. At present, most research results only aim at the preset test words and lack the universality of the entire vocabulary. Therefore, it is essential to put forward systematic solutions for all CSL vocabulary recognition.

Based on the structural characteristics of CSL, many scholars decompose it into pure structural elements for analysis and research, such as hand shape, orientation, posture, and position. Yang et al. [8] used the hand shape, orientation, position and other elements of gesture action to classify the vocabulary step by step. Although this method has high recognition rate and

**Funding:** This work was supported by the National Natural Science Foundation of China, the award

number is 62171171. This work was supported by the Natural Science Foundation of Zhejiang Province, the award number is LZ23F030005.

**Competing interests:** The authors have declared that no competing interests exist.

accuracy, it has the disadvantages of a small number of recognized words and lack of systematization. Due to the large number of gesture movements in CSL, Tigrini et al. [9] have shown that placing the collection device on the forearm or wrist is helpful in recognizing complex gesture movements. As the CSL action process has the complex characteristics of spatio-temporal information change, many scholars use multi-sensor information fusion technology to improve the accuracy of SLR. Yang et al. [10] fused image, sEMG and acceleration (ACC) sensor information to achieve high recognition rate, but the system design is extremely complex. Tian et al. [11] used the data fusion of sEMG and ACC sensors and introduced statistical language model to recognize SL, with a recognition rate of 90%. However, these studies directly input ACC eigenvalues and sEMG into the classifier, and only take ACC as the auxiliary feature of gesture state, without considering the internal correlation and dependence between gesture and movement trajectory, nor considering the spatiotemporal attributes of gesture and trajectory in the formation process of CSL. This fusion method has great limitations in the vocabulary expansion of SLR. At present, there are thousands of words are included in CSL. Identifying them one by one will not only produce a great burden of training and calculation but also complicate the recognition system.

Coupled hidden Markov model (cHMM) is a multi-stream Markov chain that describes the interaction of multiple random processes. It is highly suitable for the interactive fusion of multiple independent information streams [12]. CSL is a strong spatio-temporal action correlation process regarding gestures and trajectories. The strong timing coupling analysis ability of cHMM can effectively analyze the internal characteristics of gestures and trajectories before and after CSL. cHMM also has the advantage of analyzing the correlation and asynchronous characteristics of various source data streams [13]. When analyzing and processing the dual information flow of gesture based on sEMG and trajectory based on motion information, cHMM can not only ensure the independence of the implementation of the dual information flow algorithm, but also considers the correlation characteristics of gesture and movement trajectory at a certain SL action time. It is very suitable for the fusion of gesture and movement trajectory in SLR. On the basis of summarizing and analyzing the internal characteristics of CSL formation, this study aims to organically decompose CSL into standardized gestures and gesture-related movement trajectories, fully analyze the mode features and motion features of sEMG and motion information, output the gesture and movement trajectories related to CSL, and then integrate the two with cHMM. A systematic SLR method with more universal applicability and a richer vocabulary is proposed.

This study proposes a general method to decompose Chinese sign language into 37 standardized gestures and 18 action trajectories, and uses the sEMG, ACC and AV signals of motion information to comprehensively study the recognition of sign language words. Successfully applied to 120 common vocabulary words. This study provides a relevant foundation for the development of high real-time, high reliability and wearable sign language recognition devices.

## Materials and methods

### Participants

We enrolled 8 participants (7 men and 1 women, age: 22.1 ± 1.1 years (22–25 years)). All participants provided written informed consent, and the experimental procedures were approved by the local ethics committee of Hangzhou Dianzi University.

### Experimental preparation

A total of 8 healthy volunteers (7 males and 1 female) aged 20–30 were recruited (All subjects were informed of the specific experimental procedures and potential risks, and signed

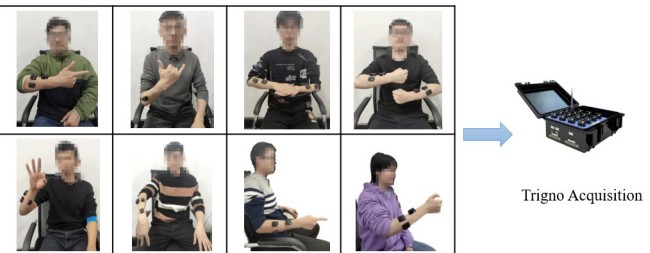

**Fig 1. Experimental process of eight volunteers.**

informed consent forms). As shown in (Fig 1), each volunteer sat on a chair and performed 120 CSL vocabulary actions respectively. The experimental acquisition system in (Fig 2) was used to record the corresponding sEMG, ACC, and AV signals. Based on these signals, 960 groups of data were collected, of which 480 groups were training samples and the other 480 groups were test samples. In the experiment, each state corresponded to a CSL vocabulary, and the likelihood probability value of the corresponding state output of each model was calculated. The composite state with the largest probability value was the target vocabulary.

The number of channels for sEMG signal acquisition and analysis influences the recognition performance, complexity and calculation of the recognition system. In general, the number of channels should be reduced as much as possible on the premise of good recognition rate, to reduce the system complexity and calculation. Therefore, according to the correlation between gestures and muscle groups, this study selects four muscle groups as signal acquisition objects: extensor carpi radialis (ECR), extensor digitorum (ED), flexor digitorum superficialis (FDS) and extensor pollicis brevis (EPB). The layout position of the four channel sEMG sensor is shown in (Fig 3).

Trigno wireless sEMG acquisition is used to build an experimental acquisition system based on sEMG and motion information. Each trigno sensor has built-in three-axis accelerometer, three-axis gyroscope and sEMG acquisition module, that can collect ACC, angular velocity (AV) signals and sEMG signals of the corresponding muscle groups in real time. The sampling frequency of the sensor is 1000Hz. The duration of a SL action is approximately 2s. During the experiment, the sensor recording sEMG signal was pasted on the surface of the corresponding muscle group of the forearm of the experimental object. The sensor recording ACC and AV signals is pasted near the wrist joint to facilitate more accurate detection of the spatial position and movement of the hand, as shown in (Fig 2).

## Chinese sign language (CSL) decomposition

CSL is developed on the basis of finger letter gesture [14], which is the research basis of CSL gesture. At present, CSL has evolved into a complex dynamic mode accompanied by limb

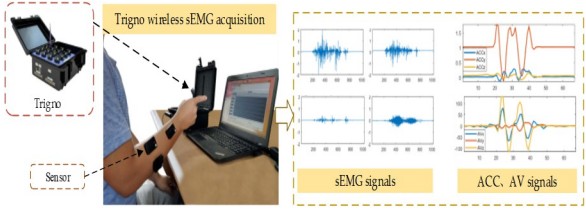

**Fig 2. Acquisition system based on sEMG and motion information.**

**Fig 3. Sensor layout position.** (a) Two muscle groups (ECR, FDS) at the front of hand side. (b) Two muscle groups (EPB, ED) at the back of hand side.

movement in the formation and change of various gestures. Therefore, the main research contents of CSL include various gestures and gesture-related movement trajectories. Therefore, this study proposes to decompose CSL into several standardized gestures and movement trajectories, organically divide a large number of CSL vocabulary recognition into gesture recognition and gesture-related movement trajectory recognition, and systematically propose an SLR scheme. Their recognition depends on sEMG and motion information of CSL action respectively. As sEMG and motion information signals have certain motion predictability [15], they are suitable for the recognition of various changing gestures and the tracking of movement trajectory. The organic combination of sEMG and motion information is also conducive to the study of the internal correlation and logic of CSL gesture and movement trajectory.

In Chinese, 30 finger letter gestures are used (Fig 4) [14, 16], of which three letter gestures are the same, with only differences in direction. The actual gestures are 27. In addition, 10 kinds of gestures (Fig 5) have been added to "Chinese Sign Language (Revised Edition)" [14]. Therefore, a total of 37 kinds of standardized CSL gestures are examined in this study. After repeated research and induction of all CSL movement trajectories, 18 kinds of regular trajectories are obtained, as shown in (Fig 6). Then, there are 19 kinds of movement states in addition to the state of action rest. According to the types of gestures, CSL is mainly divided into single hand gesture vocabulary (SHGV), double hand gesture vocabulary (DHGV) and dynamic gesture vocabulary (DGV). SHGV refers to the vocabulary expressed only by the action of the main hand (usually the right hand). DHGV refers to the vocabulary expressed by the main and auxiliary hands, and the gesture actions of these hands are the same and different. In DGV, the gesture actions in a vocabulary expression cycle are not fixed but changeable. Therefore, through the organic combination of 37 gestures and 18 movement trajectories, the number of words that can be recognized in theory is 52022, which can cover all CSL vocabulary.

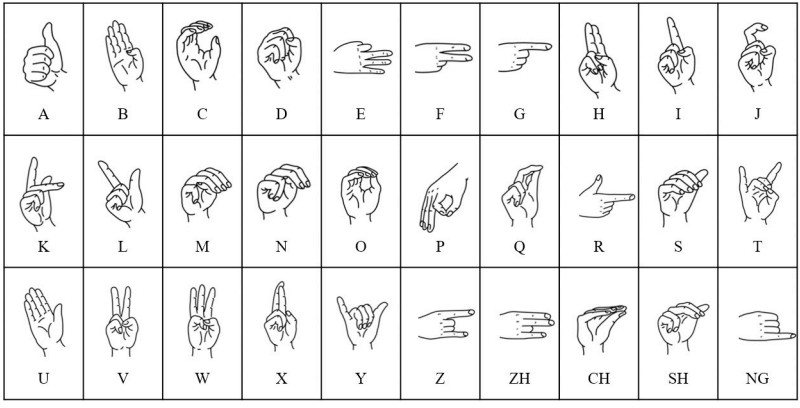

**Fig 4. Diagram of 30 finger letter gestures.**

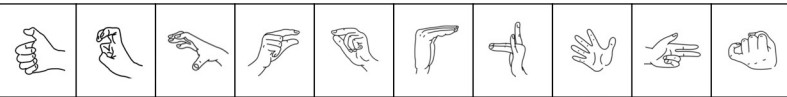

**Fig 5. Other new gestures of CSL.**

As the joint activities of more than 20 degrees of freedom of the hand are driven by specific muscle groups [17, 18], and the muscles are interrelated and coordinated, all gestures can be recognized by appropriately increasing the number and position layout of sEMG sensors. Then, 18 movement trajectories can also be detected by various motion sensors. The CSL decomposition and recognition method based on standardized gesture and movement trajectory is shown in (Fig 7). First, CSL is decomposed into standardized gestures and movement trajectories. Then, gesture recognition and movement trajectory classification based on gesture formation process are carried out respectively. Finally, the recognized CSL target vocabulary is output through cHMM fusion algorithm.

The CSL vocabulary is extensive. The scheme shown in (Fig 6) is a systematic SLR solution that can systematically summarize the recognition of all CSL vocabulary words into the recognition of 37 gestures and 18 movement trajectories. However, as the CSL vocabulary involves many uncommon words, it is neither lengthy nor complicated, nor does it necessitate identification and analysis of all words. To facilitate the analysis, this study selects 120 words involving nine gestures and eight trajectories as an example to examine the problem of SLR based on gesture and movement trajectory decomposition. Specifically, the 120 target words are presented in Table 1.

## Gesture recognition

By collecting the sEMG signals of specific muscle groups and analyzing the pattern information, nine corresponding gesture can be recognized from the sEMG signals of four muscle groups. The specific steps can be found in [19]. The definitions of the nine gestures and the corresponding CSL vocabulary are shown in (Fig 8). These nine gestures are repeated more frequently in the CSL vocabulary, which is more convenient to intuitively explain the combination form of gestures and gesture-related movement trajectories.

## Movement trajectory recognition

ACC and AV signals capture the movement trajectory information executed by SL, build the trajectory completely through the algorithm, and use the trajectory classification method to distinguish the eight movement trajectories. The specific steps can be found in [20]. (Fig 9)

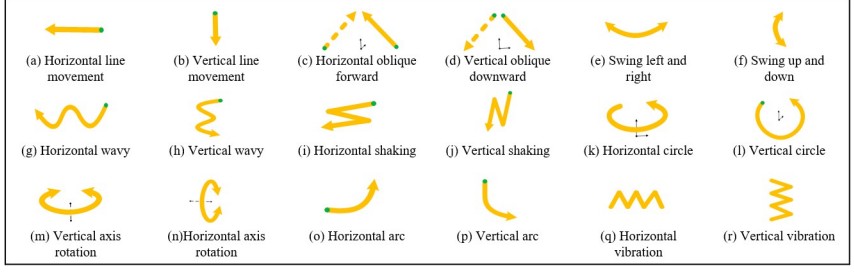

**Fig 6. Movement trajectory of CSL.**

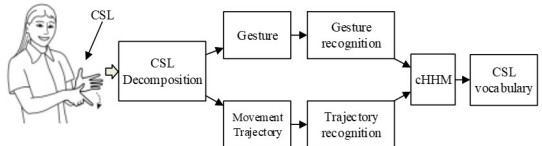

**Fig 7. Recognition scheme of CSL vocabulary.**

lists the definitions and corresponding CSL vocabulary of the eight movement trajectories in detail.

## Vocabulary segmentation based on sEMG and ACC dual-signal amplitude

A complete SL vocabulary often involves several consecutive different gesture actions, which makes the gesture segmentation algorithm based entirely on sEMG signal amplitude unsuitable for the vocabulary segmentation. In [21], the amplitude change of the sEMG signal used as the judgment basis for the start and end points of gesture action. An sEMG signal can represent the level of muscle activity. When the gesture is switched from one action to another, the corresponding muscle will relax temporarily. Therefore, the amplitude change information of the sEMG signal can be used for data segmentation of SHGV and DHGV. However, for DGV with multiple gesture combinations, judging only the sEMG signal amplitude, such as the CSL vocabulary word "clear" is not enough. (Fig 10) shows the sEMG and ACC signal activity diagram in a vocabulary cycle. In the VCM(Vertical circular arc movement) trajectory, the process of changing from FFE(Five fingers extended) to ET(Extended thumb) gesture occurs. During the gesture-switching process, the muscles relax briefly. If only the change of sEMG amplitude is used as the basis for vocabulary segmentation, the word "clear" is easy to divide into two other independent words, resulting in false recognition. In fact, the dynamic gesture process is also accompanied by the violent fluctuation of the ACC signal. According to this information, combining the amplitude changes of sEMG and ACC signals can enable more effective judgment of the start and end points of CSL activities.

**Table 1. Target CSL vocabulary.**

| Category | CSL Vocabulary |
|---|---|
| SHGV (1–45) | good (好), south (南), thank you (谢谢), you (你), me (我), two (二), eight (八), six (六), three (三), to (向), India (印度), ask (问), find (找), go (去), come on (加油), very (很), Britain (英国), phone (电话), song (歌), dumpling making (包饺子), travel (旅行), Brazil (巴西), Italy (意大利), slow (慢), hard (辛苦), know (知道), often (常常), Liu (刘), now (现在), child (孩子), just (正), Canada (加拿大), talk (说话), walk (走), expensive (贵), people (们), everyone (大家), cloud (云), halo (晕), same (同), visit (拜访), fog (雾), day (天), happiness (幸福), because (因为) |
| DHGV (46–97) | home (家), most (最), more (更), time (时间), care (照顾), marriage (结婚), sir (先生), open (开), heart (心), big (大), love (爱), invitation (邀请), contact (联系), long (长), brave (勇敢), appointment (赴约), cat (猫), warm (温暖), trousers (裤子), success (成功), special (特别), worry (担心), snow (雪), river (河), river (江), lake (湖), strip (条), laundry (洗衣服), service (效劳), cold (冷), ice (冰), friendship (友谊), understanding (认识), show (表现), today (今天), happiness (高兴), work (工作), competition (比), examine (考), grass (草), sample (样), painting (画画), sports (体育), people (人民), fry (炒), masses (群众), everything (一切), car (车), pride (高傲), truth (真), love (恋), play (玩) |
| DGV (98–120) | hear of (听说), honor (荣誉), never mind (没关系), yes (可以), parents (父母), Li (李), smooth (顺利), receiving (收货), talent (才), reception (接待), satisfaction (满意), Anhui (安徽), healthy (健康), take (拿), south Africa (南非), understanding (理解), fishing (钓鱼), handle (把), mountain climbing (爬山), PE (体检), kindergarten (幼儿园), how (怎么), clear (清) |

| Number | Diagram | Name of gesture | CSL vocabulary |
|---|---|---|---|
| 1 | | Five fingers extended (FFE) | hear of[1] (听说), to (向), warm[S] (温暖), song (歌), Brazil (巴西), handle[1] (把), PE[1] (体检), kindergarten[1] (幼儿园)，clear[1] (清) |
| 2 | | Five fingers closed (FFC) | come on (加油), contact[A] (联系), laundry[S] (洗衣服), sports[S] (体育)，handle[2] (把), how[1] (怎么) |
| 3 | | Extend thumb (ET) | good (好), smooth[2] (顺利), healthy[2] (健康), friendship[S] (友谊), Canada (加拿大), clear[2] (清) |
| 4 | | Flexion of thumb (FT) | thank you (谢谢), cloud (云), fishing[1] (钓鱼) |
| 5 | | Extend index finger(EIF) | you (你), ask (问), very (很), know (知道), talk (说话), kindergarten[2] (幼儿园), hear of[2] (听说) |
| 6 | | Extend index finger middle finger(EIFM) | two (二), walk (走), find (找), often (常常), same (同), happiness (幸福) |
| 7 | | Extend thumb and index finger(ETIF) | eight (八), Britain (英国), because (因为), Italy (意大利), brave[S] (勇敢), show (表现), sample (样), fry (炒) |
| 8 | | Extend thumb and pinkie(ETP) | phone (电话), go (去), six (六), travel (旅行), Liu (刘), visit (拜访), play[S] (玩) |
| 9 | | Flexion of thumb and index finger(FTIF) | expensive (贵), three (三), contact[B] (联系)，snow (雪)，masses[S] (群众) |

[A] represents the main hand gesture of DHGV, [B] represents the auxiliary hand gesture of DHGV, [S] represents the same gesture of the double hands of DHGV, [1] represents the first stage gesture of DGV, and [2] represents the second stage gesture of DGV. Unmarked words are SHGV.

**Fig 8. Gesture definition.**

In this study, the absolute mean sliding window (AMSW) method is used to detect the start and end points of CSL activity for the synchronization of sEMG and ACC signals. The specific steps are as follows:

- The time series of the sEMG signal with channel $k$ length $N$ is expressed as $x_k(i)$, $i = 1,2,...,$ $N$. The absolute mean value of the signal sample is:

$$MAV_k = \frac{\sum_{i=1}^{N} |x_k(i)|}{N} \qquad (1)$$

| Number | Diagram | Name of trajectory | CSL vocabulary |
|---|---|---|---|
| 1 | | Horizontal line movement(HLM) | to (向), ask (问)，find (找)，go (去)，contact (联系)，brave (勇敢), smooth (顺利) |
| 2 | | Vertical line movement(VLM) | come on (加油), very (很), Britain (英国), warm (温暖), phone (电话), snow (雪), healthy (健康) |
| 3 | | Horizontal wavy movement(HWM) | song (歌), travel (旅行), fishing (钓鱼) |
| 4 | | Vertical wavy movement(VWM) | Brazil (巴西), Italy (意大利), laundry (洗衣服) |
| 5 | | Horizontal shaking(HS) | often (常常), know (知道), Liu (刘), friendship (友谊), show (表现), handle (把) |
| 6 | | Vertical shaking(VS) | Canada (加拿大), talk (说话), expensive (贵), sample (样), walk (走), PE (体检) |
| 7 | | Horizontal circular arc movement(HCM) | same (同)，visit (拜访), fry (炒), masses (群众), sports (体育), cloud (云), kindergarten (幼儿园) |
| 8 | | Vertical circular arc movement(VCM) | happiness (幸福), because (因为), play (玩), how (怎么), clear (清), hear of (听说) |

**Fig 9. Definition of movement trajectory.**

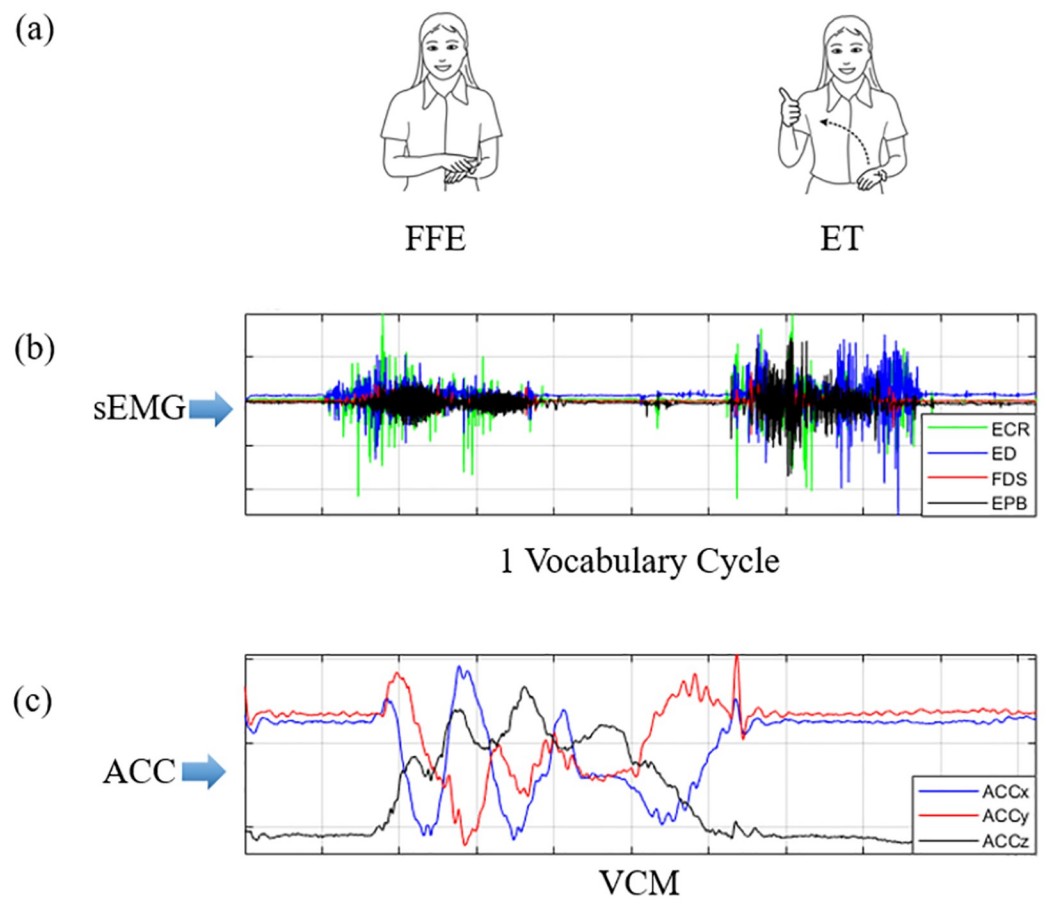

**Fig 10. Signal activity diagram of a complete vocabulary word "clear (清)".** (a) The sign language trajectory of the word "clear"(清). (b) The change of sEMG signal in one vocabulary cycle. (c) The change of ACC signal in one vocabulary cycle.

- The time series of the ACC signal with channel $l$ length $M$ is expressed as $y_l(j), j = 1,2,\ldots,M$. The absolute mean value of the signal sample is:

$$MAV_l = \frac{\sum_{j=1}^{M} |y_l(j)|}{M} \tag{2}$$

- The length of the moving window is $K$ and the step is $T$. When $K/3 \leq T \leq K/2$, better experimental results and computational efficiency are obtained. In the experiment, The moving window K = 50 and the step T = 25.

- The activity thresholds of time series sEMG and ACC signals are set to $T_{sEMG}$ and $T_{ACC}$. When $MAV_k > T_{sEMG}$ or $MAV_l > T_{ACC}$, the amplitude state is set to 1, and the signal is in the active section of the gesture action. The values of activity thresholds $T_{sEMG}$ and $T_{ACC}$ are obtained by combining the signal activity start and stop situations of 8 experimental volunteers during sign language action training.

  The CSL activity detection steps in (Fig 10) are shown in (Fig 11).

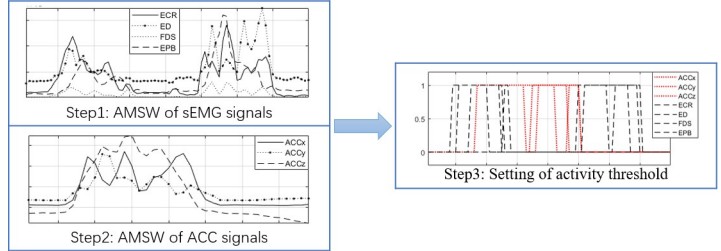

**Fig 11. The CSL activity detection steps.**

## Decision fusion of gesture and movement trajectory based on cHMM

cHMM can be regarded as a multi-chain HMM structure, and coupling conditional probability is introduced between the state sequences of each HMM [22], as shown in (Fig 12). The model consists of two HMM chains named HMMa and HMMb, which contains a hidden state sequence and an observed value sequence respectively. The number of hidden states can be set according to the actual application. The Figure shows that a certain state of each HMM chain at any time only depends on two different channel states at the previous time. This single-channel asynchronous cHMM structure retains Markov characteristics, and SLR processes the combined information of gesture and movement trajectory. The gesture and movement trajectory data are independent two channel information streams, and their modal information is also a time-dependent sequence with Markov characteristics. Therefore, the integration of SL gesture and movement trajectory using cHHM is in accordance with its internal law.

The parameters of a cHMM with two chains are described as follows [12]:

- Q: State sequence of the model. The two chains are gesture and movement trajectory sequence. Therefore, the state of the model at any time is the state combination of the two chains, which is CSL vocabulary. Note that the number of states of the $c$-th chain is $N_c$, and the number of states of the model is $\prod_{c=1}^{2} N_c$, that is, the number of CSL words. Furthermore, the $N_c$ of the $c$-th chain is $S_1^c, S_2^c, \cdots, S_N^c$, the state of the $c$-th chain at time $t$ is $q_t^c$, and the state of the model at time $t$ is $q_t = \{q_t^1, q_t^2\}$. Then the state sequence of the model is $Q = \{q_1, q_2, \cdots, q_T\} = \{(q_1^1, q_1^2), (q_2^1, q_2^2), \cdots, (q_T^1, q_T^2)\}$.

- O: Observed value sequence of the model. Similarly, the observation sequence of the model also includes the observation sequence of two chains. Note that the state of the c-th chain at time $t$ is $o_t = \{o_t^1, o_t^2\}$. Then the state sequence of the model is $O = \{o_1, o_2, \cdots, o_T\} = \{(o_1^1, o_1^2), (o_2^1, o_2^2), \cdots, (o_T^1, o_T^2)\}$.

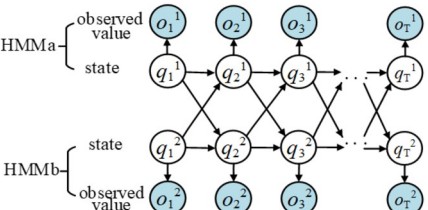

**Fig 12. Diagram of cHMM structure.**

- $\pi$: Initial state probability vector, $\pi = \{\pi_i\}$, where $\pi_i$ is the prior probability of $S_i = \{S_{i1}^1, S_{i2}^2\}$ at time $t = i$, that is,

$$\pi_i = \prod_{c=1}^{2} \pi_{ic}^c = \prod_{c=1}^{2} P(q_1^c = S_{ic}^c) \tag{3}$$

- A: State transition probability matrix, $A = \{a_{i,j}\}$, where $a_{i,j}$ is the probability of model transfer from $S_i = \{S_{i1}^1, S_{i2}^2\}$ to $S_j = \{S_{j1}^1, S_{j2}^2\}$, that is,

$$a_{i,j} = \prod_{c=1}^{2} a_{i,jc}^c = \prod_{c=1}^{2} P(q_{t+1}^c = S_{jc}^c | q_t = S_i) \tag{4}$$

Where $a_{i,jc}^c$ represents the probability that the c-th chain is in state $S_{jc}^c$ at the current time given that the model is in state $S_i = \{S_{i1}^1, S_{i2}^2\}$ at the previous time.

- B: Probability distribution of observations, $B = \{b_j(o_t)\}$, where $b_j(o_t)$ is the probability of the observed value $o_t = \{o_t^1, o_t^2\}$ when the model is in state $S_j = \{S_{j1}^1, S_{j2}^2\}$.

$$b_j(o_t) = \prod_{c=1}^{2} b_{jc}^c(o_t^c) = \prod_{c=1}^{2} P(o_t^c | q_t^c = S_{jc}^c) \tag{5}$$

Similarly, cHMM can be abbreviated as $\lambda = (\pi, A, B)$ is adjusted to maximize the probability of generating the observed value sequence. That is, a set of model parameters $\bar{\lambda}$ is found so that

$$\bar{\lambda} = \arg \max_{\lambda} P(O|\lambda) \tag{6}$$

The preceding equation is a maximum likelihood estimation problem with hidden variable Q, and the expected maximum algorithm can be used to iteratively obtain the local optimal solution. According to [11, 12], the estimated model parameter $\bar{\lambda} = (\bar{\pi}, \bar{A}, \bar{B})$ is obtained and satisfies $P(O|\bar{\lambda}) \geq P(O|\lambda)$. That is, the estimation formula always increases the probability $P(O|\lambda)$ until the local maximum is obtained. Taking the estimated model parameter $\bar{\lambda} = (\bar{\pi}, \bar{A}, \bar{B})$ as the new initial model parameter, we repeat the iterative steps of the expected maximum algorithm until the probability $P(O|\bar{\lambda})$ converges. The final model parameter is the maximum likelihood estimation of the model, that is, the obtained cHMM model. The combination state corresponding to the maximum output likelihood probability (i.e. CSL vocabulary) is the target object.

## Results and discussion

### CSL decomposition status table

In the fusion experiment of the gesture and movement trajectory information of the CSL vocabulary using the cHMM method, first, two HMM chains and implicit states of cHMM structure were defined. Nine types of gesture recognition output were defined as hidden state $(q_1^1, q_2^1, q_3^1, q_4^1, q_5^1, q_6^1, q_7^1, q_8^1, q_9^1)$ of HMMa chain. Rest state, and eight types of movement trajectory recognition output were defined as hidden state $(q_1^2, q_2^2, q_3^2, q_4^2, q_5^2, q_6^2, q_7^2, q_8^2, q_9^2)$ of the HMMb chain. The cHMM composite hidden state are combination $(q_1^1, q_2^1, q_3^1, q_4^1, q_5^1, q_6^1, q_7^1, q_8^1, q_9^1)$ and $(q_1^2, q_2^2, q_3^2, q_4^2, q_5^2, q_6^2, q_7^2, q_8^2, q_9^2)$, up to 81 types in theory.

A total of 120 CSL words are in the test vocabulary, including 45 words of SHGV, 52 words of DHGV and 23 words of DGV. Three decomposition state tables are established for the three

| | Action rest $(q_1^2)$ | HLM $(q_2^2)$ | VLM $(q_3^2)$ | HWM $(q_4^2)$ | VWM $(q_5^2)$ | HS $(q_6^2)$ | VS $(q_7^2)$ | HCM $(q_8^2)$ | VCM $(q_9^2)$ |
|---|---|---|---|---|---|---|---|---|---|
| (FFE) $(q_1^1)$ | $(q_1^1,q_1^2)$ hear of [1] | $(q_1^1,q_2^2)$ to | $(q_1^1,q_3^2)$ warm[S] | $(q_1^1,q_4^2)$ song | $(q_1^1,q_5^2)$ Brazil | $(q_1^1,q_6^2)$ handle | $(q_1^1,q_7^2)$ PE[1] | $(q_1^1,q_8^2)$ kindergarten[1] | $(q_1^1,q_9^2)$ clear[1] |
| (FFC) $(q_2^1)$ | $(q_2^1,q_1^2)$ | $(q_2^1,q_2^2)$ contact[A] | $(q_2^1,q_3^2)$ come on | $(q_2^1,q_4^2)$ | $(q_2^1,q_5^2)$ laundry[S] | $(q_2^1,q_6^2)$ handle[2] | $(q_2^1,q_7^2)$ | $(q_2^1,q_8^2)$ sports[S] | $(q_2^1,q_9^2)$ how[1] |
| (ET) $(q_3^1)$ | $(q_3^1,q_1^2)$ good | $(q_3^1,q_2^2)$ smooth[2] | $(q_3^1,q_3^2)$ heathy[2] | $(q_3^1,q_4^2)$ | $(q_3^1,q_5^2)$ | $(q_3^1,q_6^2)$ friendship[S] | $(q_3^1,q_7^2)$ Canada | $(q_3^1,q_8^2)$ | $(q_3^1,q_9^2)$ clear[2] |
| (FT) $(q_4^1)$ | $(q_4^1,q_1^2)$ thank you | $(q_4^1,q_2^2)$ | $(q_4^1,q_3^2)$ | $(q_4^1,q_4^2)$ fishing[1] | $(q_4^1,q_5^2)$ | $(q_4^1,q_6^2)$ | $(q_4^1,q_7^2)$ | $(q_4^1,q_8^2)$ cloud | $(q_4^1,q_9^2)$ |
| (EIF) $(q_5^1)$ | $(q_5^1,q_1^2)$ you | $(q_5^1,q_2^2)$ ask | $(q_5^1,q_3^2)$ very | $(q_5^1,q_4^2)$ | $(q_5^1,q_5^2)$ | $(q_5^1,q_6^2)$ know | $(q_5^1,q_7^2)$ talk | $(q_5^1,q_8^2)$ kindergarten[2] | $(q_5^1,q_9^2)$ hear of[2] |
| (EIFM) $(q_6^1)$ | $(q_6^1,q_1^2)$ tow | $(q_6^1,q_2^2)$ find | $(q_6^1,q_3^2)$ | $(q_6^1,q_4^2)$ | $(q_6^1,q_5^2)$ | $(q_6^1,q_6^2)$ often | $(q_6^1,q_7^2)$ go | $(q_6^1,q_8^2)$ same | $(q_6^1,q_9^2)$ happiness |
| (ETIF) $(q_7^1)$ | $(q_7^1,q_1^2)$ eight | $(q_7^1,q_2^2)$ brave[S] | $(q_7^1,q_3^2)$ Britain | $(q_7^1,q_4^2)$ | $(q_7^1,q_5^2)$ Italy | $(q_7^1,q_6^2)$ show[S] | $(q_7^1,q_7^2)$ sample[S] | $(q_7^1,q_8^2)$ fry[S] | $(q_7^1,q_9^2)$ because |
| (ETP) $(q_8^1)$ | $(q_8^1,q_1^2)$ six | $(q_8^1,q_2^2)$ go | $(q_8^1,q_3^2)$ phone | $(q_8^1,q_4^2)$ travel | $(q_8^1,q_5^2)$ | $(q_8^1,q_6^2)$ Liu | $(q_8^1,q_7^2)$ | $(q_8^1,q_8^2)$ visit | $(q_8^1,q_9^2)$ play[S] |
| (FTIF) $(q_9^1)$ | $(q_9^1,q_1^2)$ three | $(q_9^1,q_2^2)$ contact[B] | $(q_9^1,q_3^2)$ snow[S] | $(q_9^1,q_4^2)$ | $(q_9^1,q_5^2)$ | $(q_9^1,q_6^2)$ | $(q_9^1,q_7^2)$ expensive | $(q_9^1,q_8^2)$ people[S] | $(q_9^1,q_9^2)$ |

[A] represents the main hand gesture of DHGV, [B] represents the auxiliary hand gesture of DHGV, [S] represents the same gesture of the double hands of DHGV, [1] represents the first stage gesture of DGV, and [2] represents the second stage gesture of DGV. Unmarked words are SHGV.

**Fig 13. Status table of CSL vocabulary decomposition.**

types of vocabulary. Each vocabulary word in the table is represented as a composite state of a combination of standardized gestures and movement trajectories. After the three types of vocabulary are mapped to the state table, the recognition of the CSL vocabulary is calculated as the likelihood probability value of the cHMM state output. Due to space limitations, this study includes only the representative vocabulary in the three status tables, which are explained together (Fig 13).

## Decision fusion experiment based on cHMM

In the decision fusion experiment, first, the signal activity segment of a complete CSL vocabulary is intercepted by the vocabulary segmentation method. In the signal activity segment, three types of SHGV, DHGV and DGV are judged by the sEMG signal activity amplitude. The decision fusion experiment is conducted on the three types of vocabulary by using cHMM.

In the experiment, each state corresponds to a CSL vocabulary, and the likelihood probability value of the corresponding state output of each model is calculated. The composite state with the largest probability value is the target vocabulary. For example, in the DGV of "clear (清)", (Fig 14) is the output value of the "clear (清)" vocabulary word using the cHMM for decision fusion. (Fig 14A) shows the output value of the gesture in the first stage of the vocabulary, and (Fig 14B) presents the output value of the gesture in the second stage of the vocabulary. In (Fig 14A), Q(1,9), Q(1,8), Q(5,9), Q(5,8), Q(1,7), Q(1,6), Q(1,1), and Q(2,9) are $(q_1^1,q_9^2)$, $(q_1^1,q_8^2)$, $(q_5^1,q_9^2)$, $(q_5^1,q_8^2)$, $(q_1^1,q_7^2)$, $(q_1^1,q_6^2)$, $(q_1^1,q_1^2)$, and $(q_2^1,q_9^2)$. By looking up the decomposition status in (Fig 14), we know that the vocabulary corresponding to the output first stage gesture are "clear (清)" (first-stage gesture), "kindergarten (幼儿园)" (first-stage gesture), "hear of (听说)" (second-stage gesture), "kindergarten (幼儿园)" (second-stage gesture), "PE (体检)", "handle (把)", "hear of (听说)" (first-stage gesture), and "how (怎么)".

In (Fig 14B), Q(3,9), Q(3,3), Q(5,9), Q(4,4), Q(5,8), Q(2,6), Q(1,7), and Q(3,2) are $(q_3^1,q_9^2)$, $(q_3^1,q_3^2)$, $(q_5^1,q_9^2)$, $(q_4^1,q_4^2)$, $(q_5^1,q_8^2)$, $(q_2^1,q_6^2)$, $(q_1^1,q_7^2)$, and $(q_3^1,q_2^2)$. By looking up the decomposition status in (Fig 13), we know that the vocabulary corresponding to the output second stage

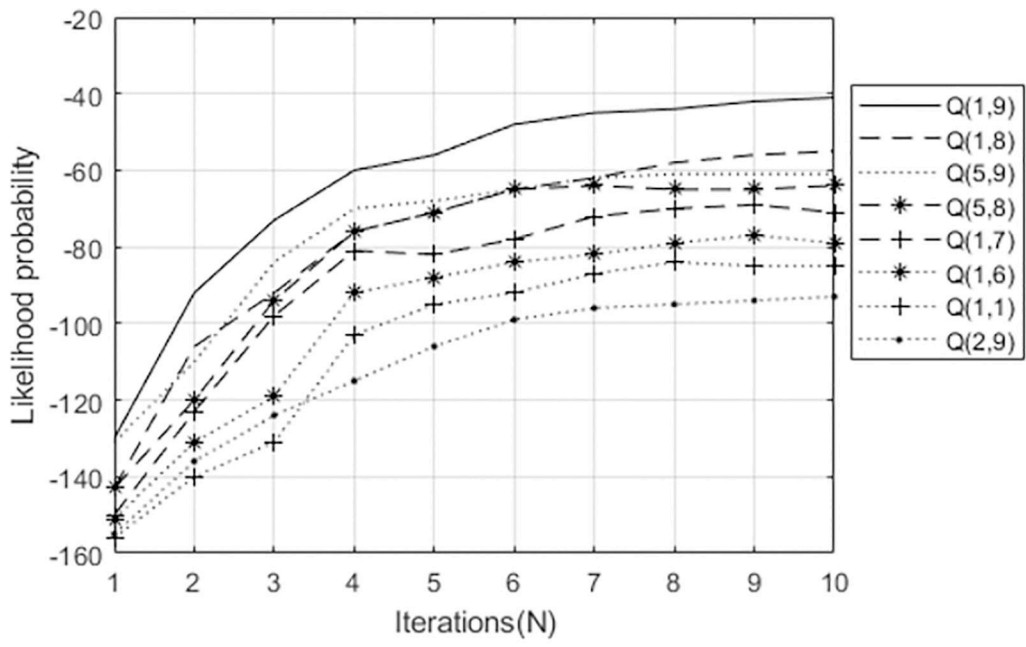

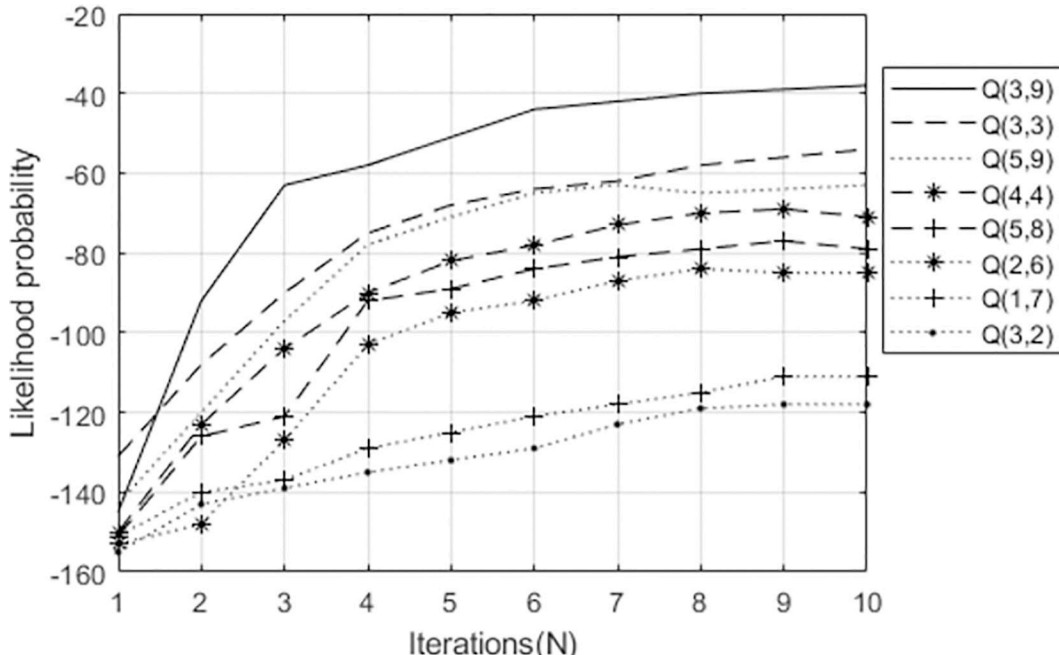

**Fig 14. Decision fusion output value of "clear (清)".** (a) First-stage gesture. (b)Second-stage gesture.

gesture are "clear (清)" (second-stage gesture), "healthy (健康)", "hear of (听说)", "fishing (钓鱼)", "kindergarten (幼儿园)" (second-stage gesture), "handle (把)", "PE (体检)", and "smooth (顺利)". The state with the largest likelihood probability value of gesture decision fusion output in the first stage and the second stage is "clear (清)", that is, "clear (清)" is the recognized target vocabulary.

**Table 2. Accuracy of different CSL vocabulary types.**

| Number | Vocabulary type | Number of words | Number of test samples | Exact quantity | Accuracy |
|---|---|---|---|---|---|
| 1 | SHGV | 45 | 180 | 166 | 92.22% |
| 2 | DHGV | 52 | 208 | 188 | 90.38% |
| 3 | DGV | 23 | 92 | 80 | 86.95% |
| 4 | Total(average) | 120 | 480 | 434 | 90.41% |

In this study, the SHGV, DHGV and DGV are trained and tested by using cHMM. Table 2 summarizes the accuracy of the three types of vocabulary, which reach 92.22%, 90.38% and 86.95% respectively. The average accuracy reaches 90.41%.

Wang et al. [23] used a three-axis ACC for the recognition of 8 custom gestures, achieving a recognition rate of 98.75%. Zhuang et al. [24] used signals from four forearm muscle groups and one palm muscle group to classify 18 Chinese Sign Language vocabulary words. The average recognition rate achieved 91.4%. The main work of [23, 24] is on gesture recognition, neglecting the action trajectories during the formation and transformation of gestures. This has resulted in a limited vocabulary for recognition. In this study, we integrate Chinese sign language gestures with action trajectory information, utilizing the cHMM multi-sensor information fusion approach to achieve the recognition of complex Chinese sign language vocabulary. The experimental results show that the decision fusion method of cHMM is effective for SL vocabulary recognition.

## Discussion

The main purpose of this article is to study gesture recognition based on the fusion of gesture information and motion trajectory information. Using the amplitude states of synchronous sEMG signal and ACC signal to determine the starting and ending points of sign language activities, continuous sign language vocabulary is segmented. Then, utilizing the independent information flow of sign language gesture and action trajectory information, as well as the inherent sequence and logical correlation, the multi-sensor information fusion method of cHMM is used to complete the recognition of sign language vocabulary. By using gesture pattern types and action trajectory types as hidden states of the cHMM chains, the output probability values of each state observation value are fused using cHMM, solving the classification problem of large vocabulary sign language recognition systems.

The vocabulary of Chinese sign language is enormous, with a total of over 5600 words. If vocabulary is identified individually or decomposed according to structural elements, it will incur a huge training burden and make the recognition system complex. Kshitij et al [1] and Heickal et al. [25] used computer vision to recognize and analyze sign language, achieving a recognition rate of 91% for 150 American sign language vocabulary. Zhou et al. [26] designed a data acquisition and recognition system for wearing on fingers using three-axis ACC, achieving an average recognition rate of 80% in 16 gesture movements. Asif et al. [2] analyzed sign language recognition based on sEMG signals and ultimately achieved a recognition rate of 95% for 11 gestures. However, the above methods only have good recognition rates for single sign language actions, and often do not have high recognition rates for continuous sign language actions. Considering the inherent order and logicality of gesture and action trajectory information in continuous sign language, this study uses the method of sEMG signal and ACC signal amplitude state to perform vocabulary segmentation, fuses sign language gesture and action trajectory to output target vocabulary, which provides convenience for continuous sign language recognition and has a positive effect on improving recognition accuracy. This study

collected sEMG and ACC data of 120 Chinese sign language vocabulary from 8 volunteers. After cHMM decision fusion, the recognition rate of SHGV and DHGV vocabulary is as high as 92.22% and 90.38%, and the recognition rate of DGV vocabulary is 86.95%. This article provides a relevant foundation for the development of Chinese sign language recognition devices with high real-time, high reliability, and wearability.

However this study also has many limitations in experiments and methods. At present, the method proposed in this study has only been tested on healthy individuals. In future work, we will cooperate with rehabilitation institutions for further experiments and research, and test this method on a group of deaf/mute patients receiving rehabilitation treatment. In addition, the target object of this article's research method is all Chinese sign language vocabulary. Currently, only 120 commonly used vocabulary libraries have been established and tested. Therefore, the expansion of the testing vocabulary library and the corresponding gesture and action trajectory decomposition table for the expanded vocabulary are also key challenges that need to be solved in the next step of research. The recognition rate of SHGV vocabulary is as high as 92.22%

## Conclusions

SLR is an important research direction in human-computer interaction. This study deeply discusses the SLR method based on sEMG and motion information fusion, and makes a beneficial attempt on the comprehensive and systematic recognition of CSL. First, according to the internal characteristics of CSL, it is divided into 37 standardized gestures and 18 movement trajectories, and all CSL vocabulary words are classified. Then, 120 commonly used target words, involving 9 gestures and 8 movement trajectories, are listed as research and test objects. Based on the acquisition system of sEMG and motion information as well as the vocabulary segmentation method of sEMG and ACC dual-signal amplitude state, the multi-sensor information decision fusion of cHMM is used to complete the recognition of CSL vocabulary. The average accuracy is 90.41%.

## Supporting information

**S1 File. Dataset.** In the file, it is reported the dataset of the experiment for each participant. (XLSX)

## Author Contributions

**Conceptualization:** Zhizeng Luo, Wenguo Li.

**Data curation:** Wenyu Li.

**Funding acquisition:** Wenyu Li.

**Investigation:** Wenyu Li.

**Methodology:** Zhizeng Luo, Wenguo Li, Xugang Xi.

**Visualization:** Wenyu Li.

**Writing – original draft:** Wenyu Li, Wenguo Li.

**Writing – review & editing:** Zhizeng Luo, Xugang Xi.

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
