## [Decision Letter · Decision Letter 0]

15 Aug 2023

PONE-D-23-22062Chinese Sign Language Recognition based on Surface Electromyography and Motion Information.PLOS ONE

Dear Dr. Li,

Thank you for submitting your manuscript to PLOS ONE. After careful consideration, we feel that it has merit but does not fully meet PLOS ONE’s publication criteria as it currently stands. Therefore, we invite you to submit a revised version of the manuscript that addresses the points raised during the review process.

We look forward to receiving your revised manuscript.

Kind regards,

Andrea Tigrini, Ph.D.

Academic Editor

PLOS ONE

Journal Requirements:

Additional Editor Comments:

Authros presented a good study on chinese sign language recognition using sEMG data. Although the applicability could be of interest, Reviewers have recognized some important major points that have to be solved otherwise leading to rejection. Authros have to clearly address all the points risedé by Reviewers.

Reviewers' comments:

Reviewer's Responses to Questions

**Comments to the Author**

1. Is the manuscript technically sound, and do the data support the conclusions?

Reviewer #1: Yes

Reviewer #2: No

2. Has the statistical analysis been performed appropriately and rigorously? 

Reviewer #1: N/A

Reviewer #2: No

3. Have the authors made all data underlying the findings in their manuscript fully available?

Reviewer #1: Yes

Reviewer #2: No

4. Is the manuscript presented in an intelligible fashion and written in standard English?

Reviewer #1: Yes

Reviewer #2: Yes

5. Review Comments to the Author

Reviewer #1: The study presented by Wenyu Li and colleagues focuses on chinese sign language recognition (SLR) based on the fusion of sEMG and motion information (through the recording of acceleration and angular velocity). 120 target words were listed and considered for the study. Then, a cHMM is employed to fuse all the recorded information and recognize words vocabulary.

Overall, the paper is clear and well written, but the methods section requires more details. Some suggestions are reported.

- Parahgraph "Introduction"

I would suggest to insert a couple of references related to the use of the sEMG signal for pattern recognition of complex task and for the development of sEMG-based human-machine interaction:

1. "Control of Hand Prostheses Using Peripheral Information", IEEE reviews in biomedical engineering, 2010

2. "Handwritten Digits Recognition From sEMG: Electrodes Location and Feature Selection", IEEE Access, 2023

- Paragraph "Materials and Methods"

Sampling frequency of the sEMG and of acceleration and angular velocity are not reported, nor the duration in time of the performed gestures. Please provide these details.

- Line 104 to 109: why are the combinations calculated as reported? Is there a specific logic from the authors or is there a reference? Please specify and refer if necessary.

- Line 127: Which is the criterion based on which the 120 words were selected? Please explain.

- Line 181-182: Acronyms (e.g. VCM, FFE, ET) should be defined in the text when used in a sentence, even if their explanation is present in a table. Please define them.

- Line 192 to 200: Is MAV the only feature considered for both sEMG and ACC signals? From the paragraph it is not clear if the MAV is calculated from the entire signal related to a word or if a segmentation process has been performed. Please clarify how the feature is extracted and if a sliding window approach is used specify also the length of the considered window, the length of the sliding and the amount of overlapping if present.

- Line 201: It is not clear the length of the moving window K and of the step T: trials have been made to find the values for K and T that give the optimal results? Which are the final values? Please add this information.

- Line 203: How the activity thresholds Tsemg and Tacc are set? Which are their values? If there are references, please report them.

- Line 210: remove "contain"

- Line 281, 291 and 320: "THGV" is not defined in the work. Is it a typing error (real term "DHGV") or is it something different? If so, please define it.

- Line 319: Article "The" missing.

If present, consider to add some references, in the discussion or conclusion section, related to similar works for a comparison of the results reported in this study.

Reviewer #2: The authors have recognized Chinese sign language using sEMG and acceleration signals and achieved the average recognition rate 90.41%. However, the novelty is not presented clearly throughout the paper. In addition, the manuscript is not written in standard format, i.e., first section of result (experimental preparation) is a part of methodology. Also, there should have a distinct discussion section describing the validation of the obtained results, limitations of the works and future directions etc. Again, the obtained performance is not compared and validated with the existing works which is very important to publish a work in a reputed journal like PLOSE ONE.

6. PLOS authors have the option to publish the peer review history of their article (what does this mean?). If published, this will include your full peer review and any attached files.

Reviewer #1: No

Reviewer #2: No

---

## [Author Response · Author response to Decision Letter 0]

15 Oct 2023

1.The manuscript format has been modified to meet the requirements of PLOS ONE.

2.I have uploaded the dataset to support information.

3.The figure in the manuscript has been converted to TIF format.

4.In the manuscript submission system, I have provided the correct Funding Information; however, the generated PDF still displays "The author(s) received no specific funding for this work" in the Financial Disclosure section.

---

## [Decision Letter · Decision Letter 1]

8 Nov 2023

PONE-D-23-22062R1Chinese Sign Language Recognition based on Surface Electromyography and Motion Information.PLOS ONE

Dear Dr. Li,

Thank you for submitting your manuscript to PLOS ONE. After careful consideration, we feel that it has merit but does not fully meet PLOS ONE’s publication criteria as it currently stands. Therefore, we invite you to submit a revised version of the manuscript that addresses the points raised during the review process.

ACADEMIC EDITOR: Authors have consistently updated the manuscript based on the previous comments. The paper is well organized and written. However, some minor points deserve to be clarified since Experts in the field still recognized minor issues that need to be carefully revised.

We look forward to receiving your revised manuscript.

Kind regards,

Andrea Tigrini, Ph.D.

Academic Editor

PLOS ONE

Journal Requirements:

Additional Editor Comments:

Paper was updated, but it still need minor points correction since a Reviewer highlightred reasonable aspects that deserves at least to be addressed.

Reviewers' comments:

Reviewer's Responses to Questions

**Comments to the Author**

1. If the authors have adequately addressed your comments raised in a previous round of review and you feel that this manuscript is now acceptable for publication, you may indicate that here to bypass the “Comments to the Author” section, enter your conflict of interest statement in the “Confidential to Editor” section, and submit your "Accept" recommendation.

Reviewer #1: All comments have been addressed

Reviewer #2: All comments have been addressed

2. Is the manuscript technically sound, and do the data support the conclusions?

Reviewer #1: Yes

Reviewer #2: Yes

3. Has the statistical analysis been performed appropriately and rigorously? 

Reviewer #1: N/A

Reviewer #2: N/A

4. Have the authors made all data underlying the findings in their manuscript fully available?

Reviewer #1: Yes

Reviewer #2: Yes

5. Is the manuscript presented in an intelligible fashion and written in standard English?

Reviewer #1: Yes

Reviewer #2: Yes

6. Review Comments to the Author

Reviewer #1: (No Response)

Reviewer #2: The authors have made a great improvement of the manuscripts. However, still there are some issues in order to meet the standard of a reputed journal.

1. The experimental results are not validated with existing literatures. So, a performance comparison section should be added.

2. In discussion, the obtained results are discussed very shortly.

3. There should be a dot after et al like Kshitij et al. [xx]

4. There should be sentence about the ethical approval and consent of the participants.

5. The major contributions of the research should be added at the end of the Introduction.

7. PLOS authors have the option to publish the peer review history of their article (what does this mean?). If published, this will include your full peer review and any attached files.

Reviewer #1: No

Reviewer #2: No

---

## [Author Response · Author response to Decision Letter 1]

20 Nov 2023

Reviewer 2:

1.The performance comparison section has been added before the discussion.

2. Enriched the discussion of the results, in the discussion section.

3.A dot has been added.

4. The ethical approval and consent of the participants has been added at the participants. 

5. The major contribution of the research has been added at the end of the introduction.

---

## [Editor Report · Decision Letter 2]

21 Nov 2023

Chinese Sign Language Recognition based on Surface Electromyography and Motion Information.

PONE-D-23-22062R2

Dear Dr. Li,

We’re pleased to inform you that your manuscript has been judged scientifically suitable for publication and will be formally accepted for publication once it meets all outstanding technical requirements.

Kind regards,

Andrea Tigrini, Ph.D.

Academic Editor

PLOS ONE

Additional Editor Comments (optional):

The manuscript was reviewed according to the comments provided by the reviewer. The quality of the manuscript fit with the PLOS ONE standards, thus it can be published.
---

## [Editor Report · Acceptance letter]

28 Nov 2023

PONE-D-23-22062R2 

Chinese Sign Language Recognition based on Surface Electromyography and Motion Information. 

Dear Dr. Li:

I'm pleased to inform you that your manuscript has been deemed suitable for publication in PLOS ONE. Congratulations! Your manuscript is now with our production department. 

Kind regards, 

on behalf of

Dr. Andrea Tigrini 

Academic Editor

PLOS ONE